

# Investigating an indirect aviation effect on mid-latitude cirrus clouds - linking lidar derived optical properties to in-situ measurements

Silke Groß[1], Tina Jurkat-Witschas[1], Qiang Li[1], Martin Wirth[1], Benedikt Urbanek[1], Martina Krämer[2,3], Ralf Weigel[3], Christiane Voigt[1,3]

[1]Deutsches Zentrum für Luft- und Raumfahrt (DLR), Institut für Physik der Atmosphäre, Wessling, 82234, Germany
[2]Forschungszentrum Jülich, Institute of Energy and Climate Research, Jülich, Germany
[3]Johannes Gutenberg-Universität, Institut für Physik der Atmosphäre, Mainz, Germany

*Correspondence to*: Silke Groß (silke.gross@dlr.de)

**Abstract.**

Aviation has a large impact on the Earth's atmosphere and climate by various processes. Line shaped contrails and contrail cirrus clouds lead to changes in the natural cirrus cloud cover, and have a major contribution to the effective radiative forcing from aviation. In addition, aviation emitted aerosols may also change the microphysical properties and, in particular, the optical properties of naturally formed cirrus clouds. Latter aerosol-cloud interactions show large differences in the estimated resulting effective radiative forcing and our understanding on how aviation induced aerosols affect cirrus cloud properties is still poor. Up to now, observations of this aviation induced aerosol effect are rare. In this study, we use combined airborne lidar and in-situ ice cloud measurements to investigate differences in the microphysical and optical properties of naturally formed cirrus clouds, which either formed under influences of aviation induced aerosol emissions or which formed under rather pristine conditions. We relate collocated lidar measurements performed aboard HALO during the ML-CIRRUS mission of the particle linear depolarization ratio with in-situ cloud probe measurements of the number and effective diameter of the ice particles. We find that those clouds, which are more affected by aviation induced aerosol emission, are characterized by larger values of the particle linear depolarization ratio. These aviation-affected cirrus clouds exhibit larger mean effective ice particle diameters connected to decreased ice particle number concentrations, than the cirrus clouds, which evolved in more pristine regions. With this study, we provide new observations of aerosol-cloud interactions, that will help to quantify related changes in the atmospheric energy budget.

## 1. Introduction

Aviation has a large impact on the Earth's radiation budget and the atmosphere (Lee et al., 2021) by various interactions; e.g. aerosols and trace gases are emitted via exhaust fumes, which directly interact with incoming and outgoing radiation (Lund et al., 2017). Line shaped contrails can form in the exhaust plume of an aircraft (Voigt et al., 2010; Burkhardt et al., 2010) and might evolve into contrail induced cirrus clouds in the aftermath (e.g. Haywood et al., 2009). A lot of research activities have


been performed over the last years to measure (Heymsfield et al., 2010; Voigt et al., 2011; Voigt et al., 2017) and understand contrails and contrail cirrus (e.g. Kärcher et al., 2015; Schuman et al., 2017), and to investigate their climate effect (Burkhardt and Kärcher, 2011; Kärcher, 2018; Bock and Burkhardt, 2019; Quaas et al., 2021). Contrails and contrail induced cirrus clouds are supposed to have the largest aviation induced impact on the Earth's radiation budget with a clear warming effect (Lee et

al., 2021). Recent studies show, that the climate impact from contrails can be reduced by burning sustainable aviation fuels with a low aromatic content (Moore et al., Nature, 2017; Burkhardt et al., 2018; Voigt et al., 2021; Bräuer et al., 2021a, b) or by climate friendly flight routing avoiding climate-sensitive air spaces (Grewe et al., 2017). Contrails can further lead to an increase in the cirrus cloud optical depth (Tesche et al., 2016) and to changes in their ice crystal effective diameter (Heywood et al., 2009).

Besides contrail formation and its effect on already existing clouds, aviation emitted aerosols might also act as ice nuclei (IN) for naturally formed clouds. The aerosols change the microphysical properties; i.e. number concentration and size of ice crystals of naturally formed cirrus clouds (Kärcher, 2017) and thus their optical and radiative effect. Model studies investigating the impact from interaction of aviation exhaust aerosols with cloud elements show large differences in the resulting effective radiative forcing. Particularly the estimates of the impact of emitted soot particles on cirrus clouds are

connected with large uncertainties. Several studies focused on the impact of aviation-exhausted soot on cirrus clouds and thus on the resulting climate effect (e.g. Hendricks et al., 2005, 2011; Liu et al., 2009; Gettelman and Chen, 2013). Large differences in the magnitude and even in the sign of the effect (Penner et al., 2009, 2018; Zhou and Penner, 2014) were reported. The uncertainties in the estimate of the climate effect of aviation emitted soot are mainly driven by the assumed efficiencies of soot particles to act as IN (Righi et al., 2021). While some laboratory studies found soot particles to be efficient IN (Möhler et al.,

2005; Hoose and Möhler, 2012), others indicate soot particles are not efficient IN (DeMott et al., 1999). In a recent laboratory study, Mahrt et al., (2020) found that soot particles would increase their efficiency to act as IN after being pre-processed within contrails. This indicates an overestimation of the soot effect in some of the model studies.

Although the understanding of the aviation's impact on the climate system has improved over the last years, many uncertainties remain, especially considering the soot effect on cirrus clouds. But, observations of an aviation induced aerosol effect on cirrus

clouds are rare. Urbanek et al. (2018) analyzed airborne lidar measurements over Europe, performed during the ML-Cirrus mission (Voigt et al., 2017), and found larger median values of the particle linear depolarization ratio (PLDR) of cirrus clouds formed in air traffic regions compared to those evolved in pristine regions. Their analysis further showed lower supersaturations for those clouds with high PLDR formed in air traffic regions, which they interpreted as a signature of more heterogeneous freezing. This measurement study is one of the first that could show traces of an indirect aerosol effect from

aviation. During the first COVID-19 curfew in spring 2020, civil aviation over Europe was reduced by up to almost 90% (www.eurocontro.int/cov19). This reduction caused a unique opportunity to study the effect of aviation on cirrus clouds. Li and Groß (2021) used spaceborne lidar measurements on board the CALIPSO satellite (Winker et al., 2010) in March and April 2020 to investigate differences in cloud occurrence and optical properties compared to former years in the same time period. They found less cirrus formation mainly for colder height levels (< -50° C) and for thinner cirrus clouds. Those findings





were interpreted as a reduction of contrails and contrail induced cloudiness due to reduced aviation. Schumann et al. (2021a, b) investigated the changes in contrail occurrence and the formation of persistent contrails by performing contrail simulations with the contrail cirrus prediction model CoCiP (Schumann et al., 2012). They found that changes in the cirrus cloud occurrence from March to August 2020, compared to the same period in 2019, was partly caused by the air traffic reduction. Theo et al., 2022 found a significant decrease in contrail cirrus cover and energy forcing in 2020, when comparing to modelled

contrail cirrus effects in the northern Atlantic flight corridor regions from 2016 to 2019.

Furthermore, a significant decrease in the mean PLDR of cirrus clouds was found in spring 2020 compared to former years (Li and Groß, 2021), which can be interpreted as a reduced, aviation induced and indirect effect on naturally formed cirrus clouds. An integrated study, using aircraft, satellite and modelling data, showed a reduction of the aerosol optical depth over Europe in May 2020 (Voigt et al., 2022), which was partly caused by the strong decline of air traffic. The restricted flight

operations furthermore led to a reduction in contrail cover and as a consequence in radiative forcing. They also found reduced effective optical depth of the cirrus clouds compared to former years connected with reduced PLDR. Looking at long-term cirrus observations using CALIPSO measurements, Li and Groß (2022) found a significant increase in the PLDR over the last years, which is clearly correlated to the increase in the number of flights over Europe. However, besides these advances in observing the change in optical properties due to the impact of aviation emitted soot, the link from aviation induced aerosol

effects on cirrus clouds to the microphysical properties of the cirrus clouds is still missing. In a recent study, Zhu et al., (2022) examined CALIPSO satellite observations during the COVID-19 lockdowns and found a significant increase in ice crystal number concentration ($N_{part}$), which they linked to an increase in homogeneous freezing due reduced aviation.

In this study we use combined lidar and in-situ measurements aboard HALO performed during the ML-CIRRUS mission (Voigt et al., 2017), to investigate differences in the microphysical properties of natural cirrus clouds formed in air traffic

regions and those formed in pristine regions. In section 2, we will present the campaign and the measurements. In section 3, we will show the results focusing first on two case studies of different cirrus cloud types and afterwards on all the cloud measurements with collocated lidar and in-situ measurements. Section 4 will discuss the results and conclude this study.

## 2. Method

### 2.1. ML-CIRRUS Campaign

The ML-CIRRUS campaign was conducted in March/April 2014 to study cirrus clouds in meteorological regimes typical for mid-latitudes. ML-CIRRUS aimed to investigate contrail cirrus, as well as to observe differences between anthropogenic and natural cirrus clouds. To achieve this goal, measurement flights with the German High Altitude and Long Range research aircraft (HALO), equipped with a combined remote sensing (including airborne lidar) and in-situ (including cloud probes) payload, were performed out of Oberpfaffenhofen. Overall, 16 flights were performed covering the whole range of the mid-

latitudes (Figure 1); from 36 to 58°N and from the Atlantic Ocean (~15°W) to Central Europe (~15°E).



During each flight, measurements at different altitudes were combined; the remote sensing instruments measured the clouds during flights well above the cirrus cloud top, while in-situ measurements were performed on several flight legs within the cirrus cloud. In this way, a combined remote sensing and in-situ data set could be sampled for different cloud regimes. An overview of the mission, the performed research flights and their main focus can be found in Voigt et al. (2017).

**Table 1: Overview of the combined in-situ and lidar research missions during ML-CIRRUS showing the mission ID, date, measurement region and scope of the mission. Listed are only those missions with combined lidar and in-situ measurements. Entries in dark blue show flight segments with cirrus clouds that have been affected by aviation emitted aerosols (according to Urbanek et al., 2018), those in light blue indicate flight segments with cirrus clouds developed in regions with no or less aviation. The flight missions indicated in bold letters are shown in detail in the case studies.**

| Mission ID | Date | Measurement Region | Scope of the mission |
|---|---|---|---|
| **M4** | **26 March 2014** | **North Atlantic flight corridor** | **Contrails and contrail cirrus** |
| M5 | 27 March 2014 | Alps, Italy, Germany | Frontal Cirrus, WCB in- and outflow |
| M6 | 29 March 2014 | France, Spain | Lee wave cirrus, WCB, jet stream divergence, convective cirrus |
| M7 | 1 April 2014 | Germany | Cirrus, contrail cirrus |
| M8 | 3 April 2014 | Germany | Frontal cirrus, WCB outflow |
| M9 | 4 April 2014 | Spain | Clean jet stream cirrus |
| M11 | 7 April 2014 | Germany | Contrail cirrus |
| **M14** | **11 April 2014** | **Great Britain** | **Frontal cirrus, WCB cirrus** |

## 2.2. WALES lidar system

The WALES (WAter vapor Lidar Experiment in Space) lidar is a combined high spectral resolution lidar (HSRL) and differential absorption lidar (DIAL) system, which was developed and built at the Institute of Atmospheric Physics of the German Aerospace Center. It measures directly the extinction coefficient at 532 nm, using the HSRL technique with a high vertical resolution of 15 m and of typically 0.2s resolution in time (Esselborn et al., 2008). Additionally, the system is equipped with polarization sensitive channels at 532 and 1064 nm. Water vapor concentration is measured by simultaneously emitting laser pulses at three online and one offline wavelength in the water vapor absorption band around 935 nm (Wirth et al., 2009). The overlapping range contributions of the three online wavelengths provide the full information of the water vapor profile from just below the aircraft down to ground level. For determining the particle linear polarization ratio (PLDR), we apply the ±45° calibration method (Freudenthaler et al., 2009), and thus achieve an absolute accuracy of 5 percentage points at typical cirrus PLDR values.

To assure that high altitude aerosol residuals and liquid or mixed phase clouds are excluded from our study, we restrict the considered data to measurements of temperature ranges below 235 K and of a backscatter ratio (R) above a threshold of R=3.





This threshold was determined by carefully investigating all flights with lofted aerosol layers (Urbanek et al., 2018). Sensitivity studies showed, that the further analysis only weakly depend on the chosen R value within a range from R=2 to R=25.

### 2.3. In-situ Instrumentation

NIXE-CAPS (Novel Ice Experiment–Cloud Aerosol and Precipitation Spectrometer; Krämer et al., 2016; Costa et al., 2017) is a combination probe that integrates two techniques for measuring the particle size distribution (PSD): the PSD of particles 0.6

to 50 µm in diameter is measured with NIXE's Cloud and Aerosol Spectrometer (NIXE-CAS) using light scattered from individual particles that pass through a focused laser beam. For measurements of particles 15–937 µm in diameter, NIXE's Cloud Imaging Probe (NIXE-CIP-grey threshold), which utilizes the optical array probe (OAP) technique, is used. Using the data analysis routines collected in the NIXE-Lib, the PSDs of both instruments are analyzed simultaneously, whereby various error analyzes and corrections are applied, including a correction of possible shattering of large ice crystals at the inlet tips.

Particle number concentrations ($N_{par}$) and effective diameter ($D_{eff}$) are calculated using a composite of particle size distributions from three cloud probes, applying scattering detectors and light attenuation on optical arrays. Small particles in the size range from 3 to 50 µm were detected by the CAS-DLR (Voigt et al. 2017, Kleine et al. 2018). The data have been grouped into 16 size bins, assuming rotationally symmetrical ellipsoids of random orientation with aspect ratios of 0.75, to avoid Mie-ambiguities in the scattering signals. Larger particles were detected by the greyscale Cloud Imaging Probe (CIPg-UniM) in 14

size bins covering diameters from 25 µm to 1 mm (Molleker et al., 2014; Klingebiel et al., 2015; Mei et al., 2020) as part of the CCP (Cloud combination probe) and a Precipitation Imaging Probe (PIP) instrument for characterizing precipitating cloud elements and hydormeteors of diameters from 100 µm to 6400 µm (Weigel et al., 2016). Maximum dimension diameters were derived from 2D images and number concentrations were corrected for compression effects according to Weigel et al. 2016. The effective diameter is calculated according to Schumann et al. 2010. The data have been averaged over 5 s intervals. For

the flight of 7 March 2014, only data from the NIXE-CAPS instrument are available. Comparison of the data sets for all other days showed a good agreement.

### 3. Results

During the ML-CIRRUS campaign, eight missions were performed which provide coordinated lidar and in-situ measurements (Table 1). The results of those coordinated measurements will be presented in the following. Two case studies show the

differences in the optical and microphysical properties of similar cirrus cloud types; 1) from a region with elevated background aerosols due to aircraft emissions, and 2) from an air mass comparatively unaffected by air traffic. One of the two case studies shows measurements of cirrus clouds strongly affected by embedded contrails and the other of a warm conveyor belt cirrus. Those cases represent two of the main cirrus types in the European mid-latitudes. In a next step, we investigate the overall distribution of optical and microphysical properties of the observed cirrus clouds.





### 3.1. Case study – Contrail Cirrus

The first case study we choose for the comparison is a cirrus case with embedded fresh contrails (Wang et al., 2022). For the clouds observed on 26 March 2014 and 7 April 2014, the contrail cirrus prediction model (CoCiP; Schumann 2012) indicated a large amount of embedded fresh contrails within the cirrus cloud (Urbanek, 2019). Back-trajectory analysis (Urbanek et al., 2018) indicate the position where the cloud formation had occurred on 26 March over the North Atlantic with enhanced background aerosol due to aircraft exhaust. The cirrus cloud on 7 April evolved further south over the Atlantic Ocean, in an area that is much less affected by aviation exhaust (Stettler et al., 2013).

The time-height cross-sections of the particle linear depolarization ratio (PLDR) of the two cirrus clouds with embedded fresh contrails is shown in Figure 5 ((a) and (b)) along with the density distributions ((c) and (d)) of the PLDR. Both cirrus clouds are in approximately the same temperature and height range, so they offer commonalities on the basis of which a comparison would be appropriate. The values of the PLDR of the two cirrus clouds is quite different. The cirrus on 26 March 2014 shows larger values of the PLDR than the cirrus cloud on 7 April 2014. For the first one, we find values up to 0.6, while the PLDR of the cloud on the 7 April barely exceeds 0.45. The mode of the PLDR distribution within the cirrus cloud on 26 April is about 0.54, its median ranks about 0.52. In contrast, the mode and median of the PLDR distribution of the cirrus cloud on 7 April 2014 is much lower at values of about 0.34 and 0.29, respectively. Both cirrus clouds show a large number of embedded contrails and are still different with respect to their PLDR. Thus, the freshly embedded contrails cannot be interpreted as a cause for the significant differences in the PLDR.

In-situ measurements were performed within the cirrus cloud at an altitude of approximately 11 km and show mean relative humidity with respect to ice (RHi) and temperature values of about 100% and about 210 to 212 K, respectively, for both clouds. So, any differences due to changed measurement conditions within the cloud are not expected. For 26 March 2014, combined CAS-DLR/CIPg-UniM and NIXE-CAPS data is available. For 7 March 2014 CIPg-UniM data are missing. Thus, only the NIXE-CAPS data is used for that day. As CAS-DLR/CIP-UniM and NIXE-CAPS data show agreement on all other days, this has no effect on the result. The distributions of the derived $D_{eff}$ are narrow for both cirrus cases, with the main values below 100 µm (Figure 3) potentially due to the high number of embedded contrails. The median $D_{eff}$ value for the cirrus cloud measured on 26 March of about 60.8 µm is slightly larger than the median value of approx. 54.6 µm measured on 7 April 2014. Differences are also found for the measured $N_{par}$ for the two cloud cases. The distribution for the cirrus cloud evolving in regions with large amount of aviation exhaust (26 March) shows a median value of 0.04 cm$^{-3}$. In contrast, the distribution of $N_{par}$ on 7 April is broader and shows a larger median value of about 0.07 cm$^{-3}$. In summary, the case study shows larger median effective diameter and lower number concentration for the aviation impacted cloud with the high particle linear depolarization ratio mode. This result is discussed in more detail in the conclusion.



### 3.2. Case study – Warm Conveyor Belt Cirrus

In a second case study, we compare the optical and microphysical properties of warm conveyor belt (WCB) cirrus. WCBs are a typical cloud / flow structure of the mid-latitudes leading to increased precipitation (Eckhardt et al., 2004). A warm conveyor belt (WCB) is characterized by warm humid air that is lifted quickly (~ 10 cm/s; Browning, 1971) from the lower troposphere to higher levels over several kilometers. During the lifting process, liquid clouds form and freeze. This leads eventually to ice clouds. During ML-CIRRUS, we were able to observe and probe four cases of WCB cirrus (see Table 1). For this study, we choose the clouds observed on 27 March 2014 and on 11 April 2014. Using the cirrus lifetime classification method presented by Urbanek et al. (2017), we can show that both observed clouds are approximately in the same stage of lifetime. Both clouds are in a well-developed stage with a tendency to dissolve (Figure 4). The classification scheme is based on previous studies using differences of the RHi distribution in clouds at different stages of evolution (Groß et al., 2014) Only about 0.4-0.5% of each of the two clouds are in the nucleation mode, and about 30% are in the deposition mode. The largest part of both clouds (~70%), however, is in the sublimation mode. Thus, the similar stage of development of two clouds suggests a comparison of the cloud properties, as any effect that coulf result from different stages of the clouds' maturity is ruled out. Trajectory analyses for the two cloud cases show, that the first one evolved in rather clean, from aviation exhaust unaffected situations over northern Africa / the Mediterranean, while the nucleation process for the second one took place over the north Atlantic region, highly affected by air-traffic exhaust (Figure 1).

Figure 5 shows the time-height cross-sections ((a) and (b)) and density distributions ((c) and (d)) of the particle linear depolarization ratio (PLDR). While the PLDR of the WCB cirrus on 27 March 2014 does not exceed values of 0.55, the PLDR of the WCB cirrus on 11 April 2014 shows values as high as 0.7 at the top and in the lower part of the cloud. Taking all the measurement points within the observed WCB cirrus on 27 March into consideration, the overall distribution of PLDR has its maximum at a value of 0.41; the median of the distribution is at 0.4. In contrast, the distribution of the PLDR of the WCB cirrus on 11 April has its maximum at 0.5 and its median at a value of 0.48. As we find the differences in the measured PLDR for both WCB cases, and as their stage of evolution is approximately the same, we suggest that neither the cirrus cloud type nor its stage of evolution has an effect on the differences in the PLDR.

In-situ measurements provide information of the relative humidity with respect to ice (RHi) and on the temperature along the flight path (Kaufmann et al., 2019). On 27 March 2014, mean values along the in-situ flight track of 105% (std 18%) and 223 K (std 14 K) were measured. The mean value of the RHi and temperature along the in-situ flight track on 11 April 2014 were 104% (std 13%) and 232 K (std 3 K). The distribution of $D_{eff}$ (Figure 6; left) for the WCB cirrus on 27 March 2014 has a narrow mono-modal structure with its maximum between about 80 to 110 µm. In contrast, $D_{eff}$ within the WCB cirrus on 11 April shows a broader distribution with its maximum at about 200 µm and a second smaller mode at about 100-120 µm. The overall mean (median) values of $D_{eff}$ for the WCB cirrus on 27 March and 11 April are 94.02 µm (98.58 µm) and 189.77 µm (187.05 µm), respectively. The distribution of $N_{par}$ shows now significant differences for the two WCB cirrus cases. Both





distributions show a skewness towards smaller values and a median at about 0.03 cm⁻³. The comparison of the derived $D_{eff}$

distribution indicates that the cirrus, which has formed in regions highly affected by air traffic, shows larger ice particles.

Unfortunately, the in-situ flight tracks were not planned with respect to temperature regimes. But the temperature might have

an impact on the comparability of the optical and microphysical properties of the cloud. This will be discussed in more detail

in Section 3.3.

### 3.3. In-situ data of all flight missions

Looking at the overall distribution of the derived effective diameter and ice particle number concentration (Figure 7) for all

cirrus clouds with coordinated lidar and in-situ measurements during ML-CIRRUS (Table 1), we do not find a clear

dependence of size and number of ice particles on the particle linear depolarization ratio (PLDR). The median value of $D_{eff}$

(88.6 µm) for the clouds with lower PLDR is even slightly higher compared to a median $D_{eff}$ of 78.0 µm for the high

depolarization mode clouds. The corresponding median values of the ice particle number concentration within this temperature

range are quite similar with about 0.05 cm⁻³ for the high and low mode depolarization cirrus. However, for this analysis the

whole temperature range from 206 to 238 K is considered. But similar to its impact on the PLDR (Urbanek et al., 2018) the

temperature is also correlated with the mean/median $D_{eff}$ and $N_{par}$ (Figure 8) showing lower values for $D_{eff}$ and higher values

for $N_{par}$ at the coldest temperature. Looking at Table 1, one can see that coordinated in-situ and lidar measurements were

mainly performed in cirrus clouds formed in rather pristine airmasses. Furthermore, for the temperature range below 210 K,

those in-situ measurements are dominating the available data, that occurred in cirrus clouds exhibiting high PLDR. In contrast,

in the temperature range above 215 K the in-situ measurements, that occurred in cirrus clouds in the low PLDR mode, are

dominating the data availability. This might affect the overall results. Thus, we compare the derived $D_{eff}$ and $N_{par}$ in a

temperature range between 210 K and 215 K (Figure 7; right), where in-situ measurements of both cirrus cloud types are about

equally available. Looking at this temperature regime (210-215 K) one can see slight differences between the two cirrus cloud

classes. The main values of $D_{eff}$ for both cloud types are between about 25 µm and 100 µm. But the median value of the $D_{eff}$

distribution (50.7 µm) within low mode PLDR cirrus clouds is slightly smaller than the median $D_{eff}$ value for the in-situ

measurements within high mode PLDR cirrus clouds (61.4 µm). The corresponding distributions of $N_{par}$ show median values

of about 0.05 for the high mode PLDR clouds and of 0.11 for the low mode PLDR clouds.


In a next step, we analyze the temperature dependence of the measured effective diameter and ice particle number

concentration. For this evaluation, we use the temperature range from 208 to 217 K, as only in this temperature range we do

have sufficient data for both cloud types. We derive the distributions of $D_{eff}$ and $N_{par}$ in 1 K steps (Figure 8). As already seen

for the overall distribution, we do not find clear differences of $D_{eff}$ for the high mode PLDR clouds and the low mode PLDR

clouds for most temperature steps. But again, the distributions are dominated by high mode PLDR clouds in the colder regions

and by low mode PLDR clouds in the warmer regions. And, one can see a tendency towards larger $D_{eff}$ with warmer



temperatures. For the high mode PLDR clouds the median $D_{eff}$ is 22.9 µm at a temperature of 208 K and of 65.4 µm for a temperature of 216 K. The corresponding values for the low mode PLDR clouds are 28.8 µm and 67.2 µm, respectively.

The corresponding distributions of the ice particle number concentration show, that $N_{par}$ is larger for the low mode PLDR cirrus clouds than for the high mode cirrus clouds through all considered temperature ranges. The median value of $N_{par}$ is 0.47 cm$^{-3}$ for the low mode PLDR cloud and 0.22 cm$^{-3}$ for the high mode PLDR cloud at a temperature of 208 K, and of about 0.01 cm$^{-3}$ for the low mode PLDR cirrus and about 0.07 cm$^{-1}$ for the high mode PLDR cirrus at a temperature of 215 K. However, as can be seen from Table 1, these last results have to be treated with care, as we have a larger number of high mode

cirrus cloud cases in the lower temperature ranges and a dominance of low mode cirrus cloud cases in the higher temperature ranges. Looking only at those temperatures with approximately the same contribution from both cloud types (210-215 K), the distribution of $D_{eff}$ and $N_{par}$ mainly show slightly smaller median particle diameters for the low mode PLDR clouds with corresponding higher median values of the ice particle number concentration.

**Discussion and Conclusion**

In our study, we used the same method and measurements as Urbanek et al (2018), who showed for the first time a difference in the optical properties (i.e. the particle linear depolarization ratio) for cirrus clouds that formed in regions with large aviation induced emissions (having higher values of the PLDR) and those that formed in less affected regions (lower values of the PLDR). We connected the lidar measurements with collocated in-situ measurements of ice particle size and ice particle number concentration from cloud combination probes on HALO, where available.

We found, that for those temperature regimes, where we have a sufficient contribution of both cloud types, high PLDR mode clouds show lower ice particle number concentrations with larger effective diameters compared to low PLDR mode clouds. That is an indication for more heterogeneous freezing due to aviation induced emissions, as homogeneous nucleation is expected to be suppressed by heterogeneous nucleation (DeMott et al., 1997; Gierens, 2003). Homogeneous freezing might still occur sometime after the heterogeneous process according to Spichtinger and Cziczo (2010). They further showed, that

heterogeneous freezing takes place at lower RHi. In their study, Urbanek et al., (2018) investigated the distribution of RHi inside high mode PLDR clouds and low mode PLDR clouds and found differences in the supersaturation with larger values for the low mode PLDR clouds. These higher values can thus be interpreted such that homogeneous freezing plays a larger role in the low mode PLDR clouds. Homogeneous freezing is expected to produce high ice crystal number concentration and small crystal sizes (Kärcher et la., 2006, Spichtinger and Cziczo, 2010; Krämer et al., 2016). This was found during the

COVID-19 curfew with strongly reduced aviation (Zhu et al., 2022) and thus decreased number of ice nucleating particles (INP). In the presence of solid aerosol particles that act as INP, ice crystals form at lower supersaturation. The INP are comparably less numerous than the small aerosol solution droplets causing homogeneous freezing. Thus, the available water vapor deposits on a smaller number of ice crystals but grow to larger sizes and potentially also more complex ice crystals (Schnaiter et la., 2016). The availability of INP and thus heterogeneous freezing processes lead furthermore to lower ice particle



number concentration in a subsequent homogeneous freezing process compared to pure homogenous freezing (Spichtinger and Cziczo, 2010; Krämer et al., 2016). This effect is stronger the more INP are available.

The differences of $D_{eff}$ and $N_{par}$ for the high and low mode PLDR clouds, with larger $N_{par}$ but slightly smaller $D_{eff}$ for the low mode PLDR clouds, can thus also be interpreted as traces of more frequent heterogeneous freezing in the high mode PLDR

clouds. Similar results were also found in a recent study investigating the changes in ice crystal number concentration during the COVID-19 caused air traffic closure (Zhu et al., 2022). They found a reduction in the ice crystal number concentration during that period and interpreted it as an increase in homogeneous freezing as soot from aircraft exhaust was reduced. Li and Groß (2021) investigated the optical properties of cirrus clouds over the European mid-latitudes and found a reduction in the PLDR of cirrus clouds during the COVID-19 lockdown in spring 2020. However, number concentration and crystal size are

not expected to be the only microphysical properties to affect the measured particle linear depolarization ratio, directly. It also depends on the crystal habit or surface roughness, thus on the complexity of the particles. Conditions during the nucleation process (e.g. temperature, relative humidity) impact the ice crystal shape (Bailey and Hallett, 2009). More heterogeneous freezing at lower supersaturation (Urbanek et al., 2018) and warmer temperature regimes (Kanitz et al., 2011) as expected for the high mode PLDR might lead to changes in the ice crystal complexity. Larger particles with more complex structure where

found e.g. from balloon-borne measurements of cirrus clouds (Heymsflield, 2003) under such conditions. In this study, we also find a temperature dependence of $D_{eff}$ and $N_{par}$ with larger $D_{eff}$ and lower $N_{par}$ for the warmer cloud temperature range.
The ML-CIRRUS campaign was not designed to investigate an indirect aviation effect of cirrus clouds, and aerosols in the regions of cirrus cloud formation were not explored in detail. Still, it was possible to derive important information of the data. Furthermore, the original focus of ML-CIRRUS was not on a sufficient collocation of lidar and in-situ measurements for both

cloud types. Therefore, additional measurements were performed during the CIRRUS-HL mission in 2021 and the flight planning for the CIRRUS-HL campaign learned from our experiences and the collocation of lidar and in-situ measurements was particularly improved. Also, the sampling of aerosol properties in the region of the cloud formation and evolution was a focus of CIRRUS-HL. In the following, we expect that data from the CIRRUS-HL mission will address more of the open questions related to the impact of aviation on ice cloud properties.

**Data availability**

The data used in this study are available at the HALO database (halo-db.pa.op.dlr.de).



**Author contributions**

SG performed the lidar measurements, CV, TJW, MK, and RW performed the in-situ measurements during ML-CIRRUS. MW provided the basic analysis of the lidar data. MK, TJW provided the basic analysis of the in-situ data. BU, QL, SG
performed the analyzes in this study. SG wrote the manuscript. All authors discussed the data and findings.

**Competing interests**

The authors declare that they have no conflict of interest.

**Acknowledgements**

We thank Marcus Klingebiel (Universität Leipzig, Germany) and Sergej Molleker (MPIC, Germany) for performing and
providing the CIPg, CCP and PIP data during. We are grateful to Tiziana Bräuer (DLR; Germany) for helpful comments on the manuscript. ML-CIRRUS was mainly funded by the Deutsches Zentrum für Luft- und Raumfahrt (DLR), and the Deutsche Forschungsgesellschaft (DFG) within the SPP1294-HALO under contract VO1504/7-1, and by the Helmholtz Association under contract W2/W3-60. This study was funded by CRC project TRR 301 under Project-ID 428312742, and DLR internal funding within the MABAK project.

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



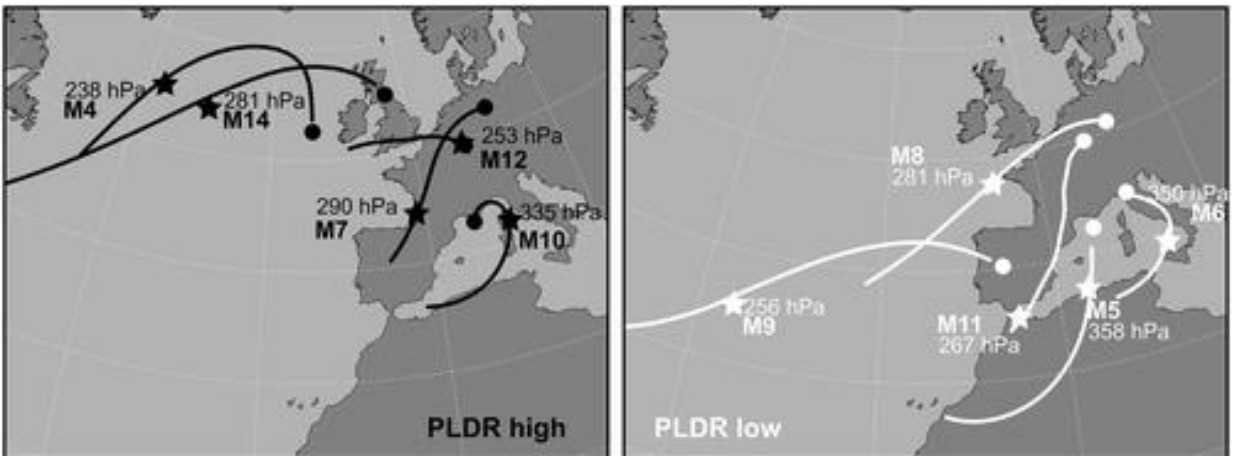

**Figure 1: Location of the of cirrus clouds measured by lidar during ML-CIRRUS (dots), their history as derived from backward trajectories (solid lines), and the most likely regions of cirrus cloud formation (stars) for PLDR high mode clouds (left) and for PLDR low mode clouds (right). The figure is taken from Urbanek et al, 2018 (Figure 3). Measurements during missions M4, M5, M6, M7, M8, M9, M11, and M14 are used in this analysis.**

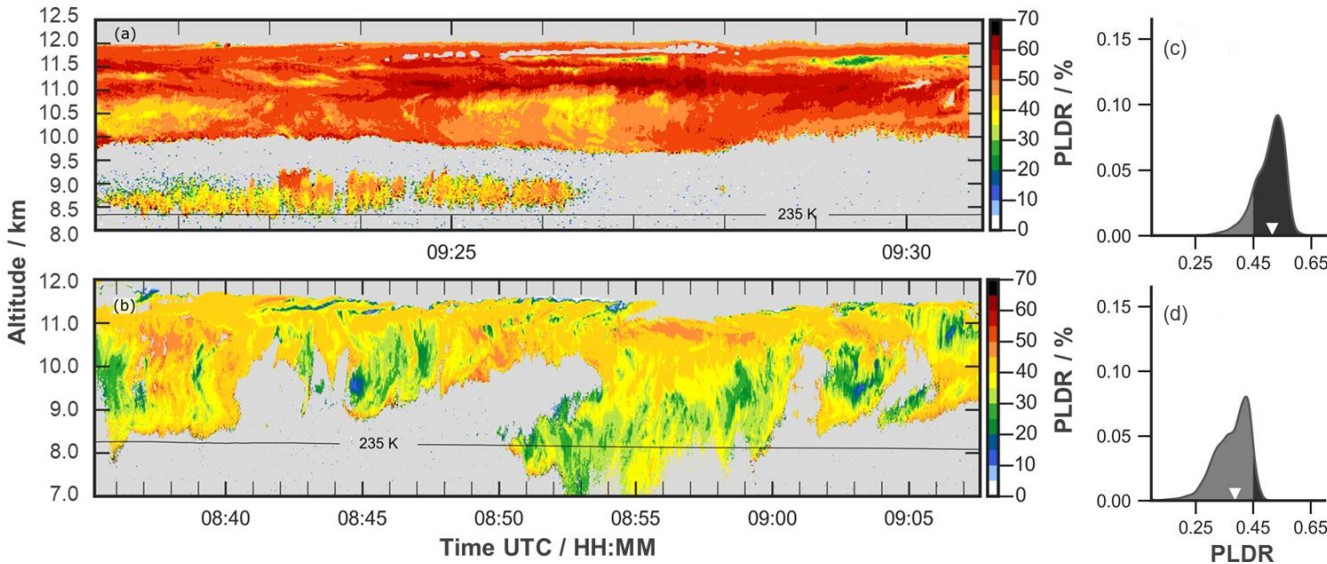

**Figure 2: Time-height cross-section of the particle linear depolarization ratio (PLDR) for the contrail affected cirrus cloud cases on 26 March (a) and 07 April 2014 (b), and the frequency distribution of the measured PLDR (c, d) of the cloud parts at temperature regions below 235 K. The color coding of the PLDR distribution (c and d) marks the threshold defined by Urbanek et al., 2018 for high (dark grey) and low (light grey) PLDR values.**





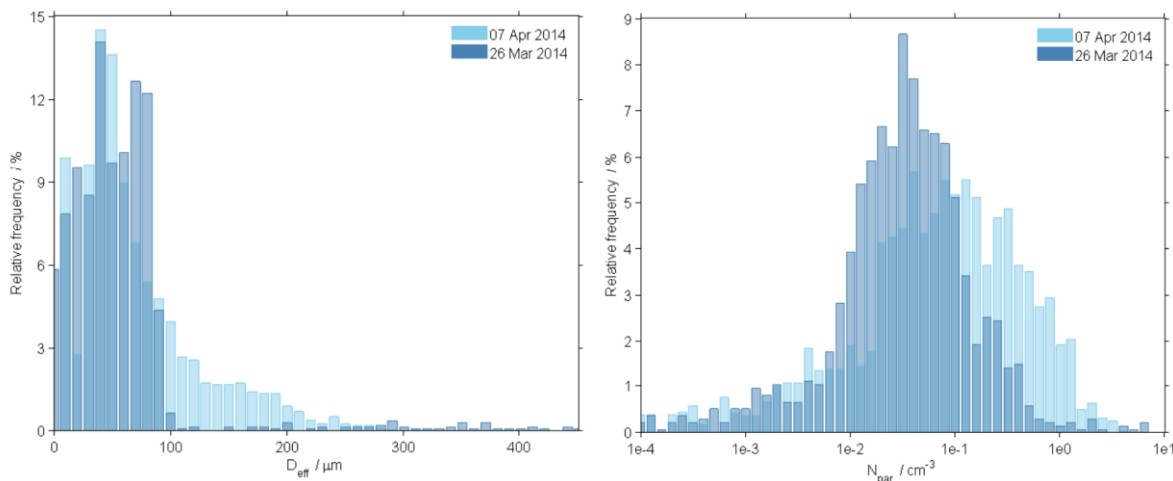

**Figure 3: Relative distribution of the retrieved effective diameter Deff (left) and ice particle number concentration Npar (right) for the contrail affected cirrus cloud cases on 26 March (dark blue) and 7 April (light blue) 2014 from in-situ measurements during ML-CIRRUS.**

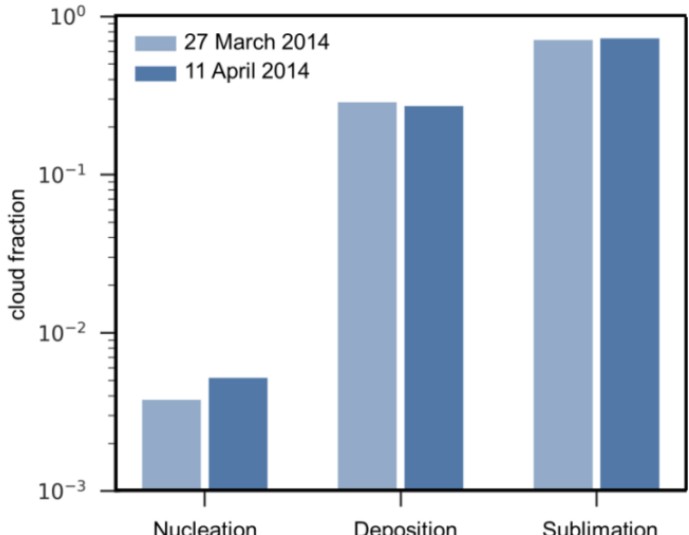

**Figure 4: Fraction of the clouds on 27 March 2014 (light blue) and 11 April 2014 (blue) in the nucleation (heterogeneous and homogeneous), deposition and sublimation mode.**

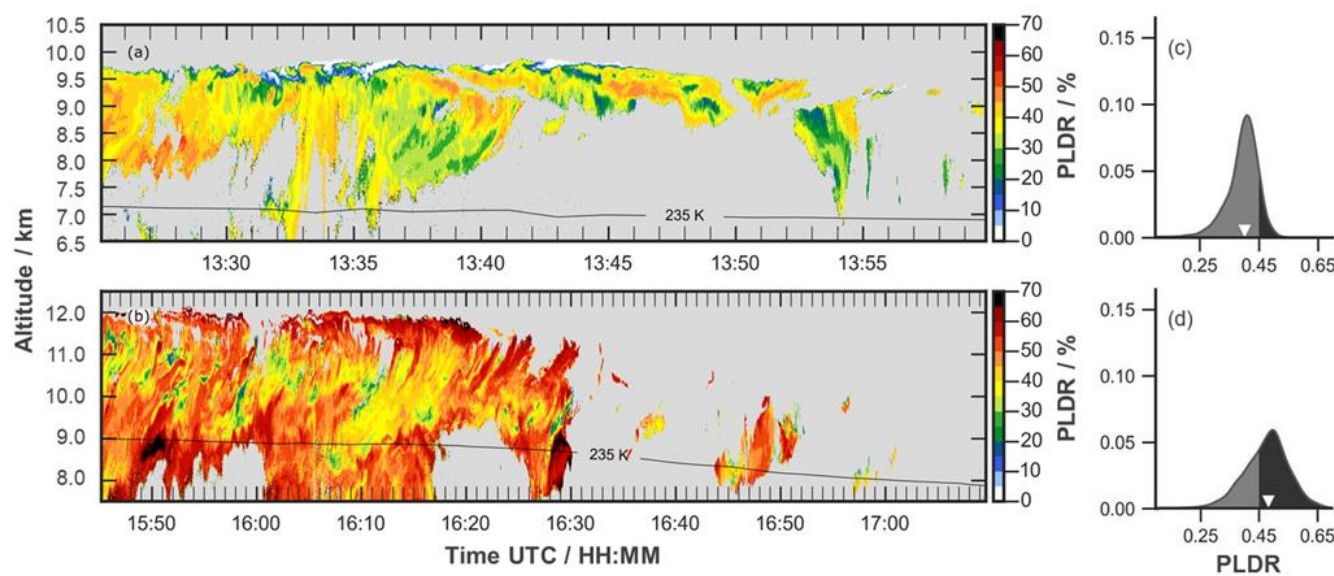

**Figure 5: Time-height cross-section of particle linear depolarization ratio (PLDR) for the WCB cirrus observed on 27 March (a) and 11 April 2014 (b), and the frequency distribution of the measured PLDR (c, d) of the cloud parts at height regions below 235 K.**
525 **The color coding of the PLDR distribution marks the threshold defined by Urbanek et al., 2018 for clouds characterized by high (dark grey) and low (light grey) PLDR values.**

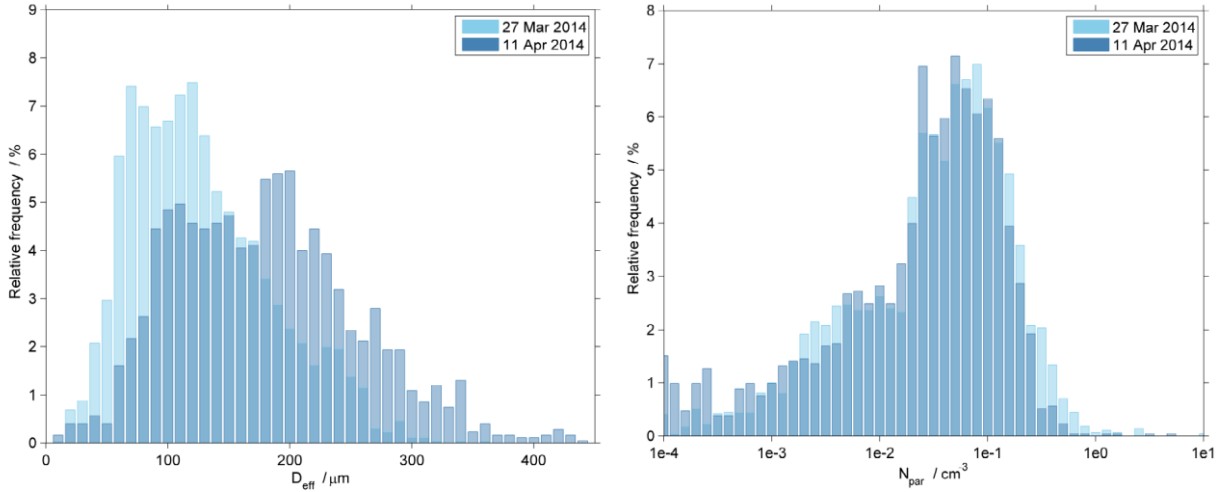

**Figure 6: Relative frequency of the derived effective diameter (left) and ice number concentration (right) for the warm conveyor**
530 **belt cases on 27 March (light blue) and 11 April (dark blue) 2014.**





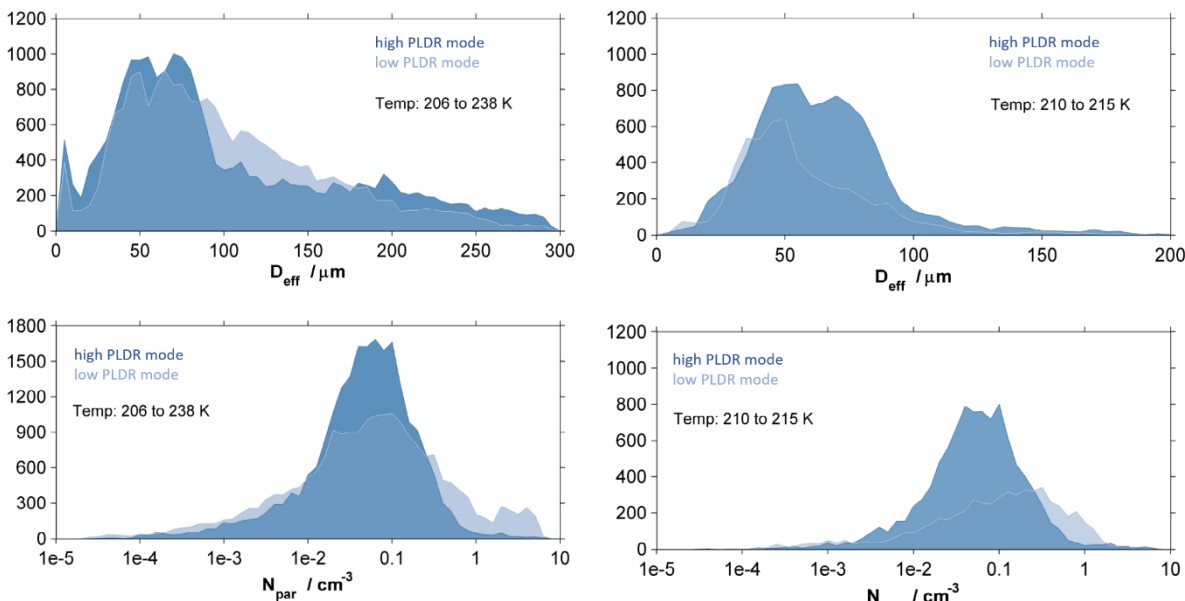

**Figure 7: Probability density function of the measured ice particle effective diameter (upper plots) and ice number concentration (lower plots) derived from CAS-DLR/CIPg-UniM and NIXE-CAPS data (where available) for all cirrus clouds during ML-CIRRUS, where coordinated lidar and in-situ measurements were available (Table 1).  The light blue color indicates measurements in low PLDR mode clouds and the one in dark blue show measurements for high PLDR mode clouds. The distributions on the left use measurements of all in-situ data within cirrus clouds. For the distributions on the right only measurements in a temperature regime from 210 K to 215 K are used, as only in this temperature regime approximately the same number of measurements within high and low PLDR mode clouds are available.**





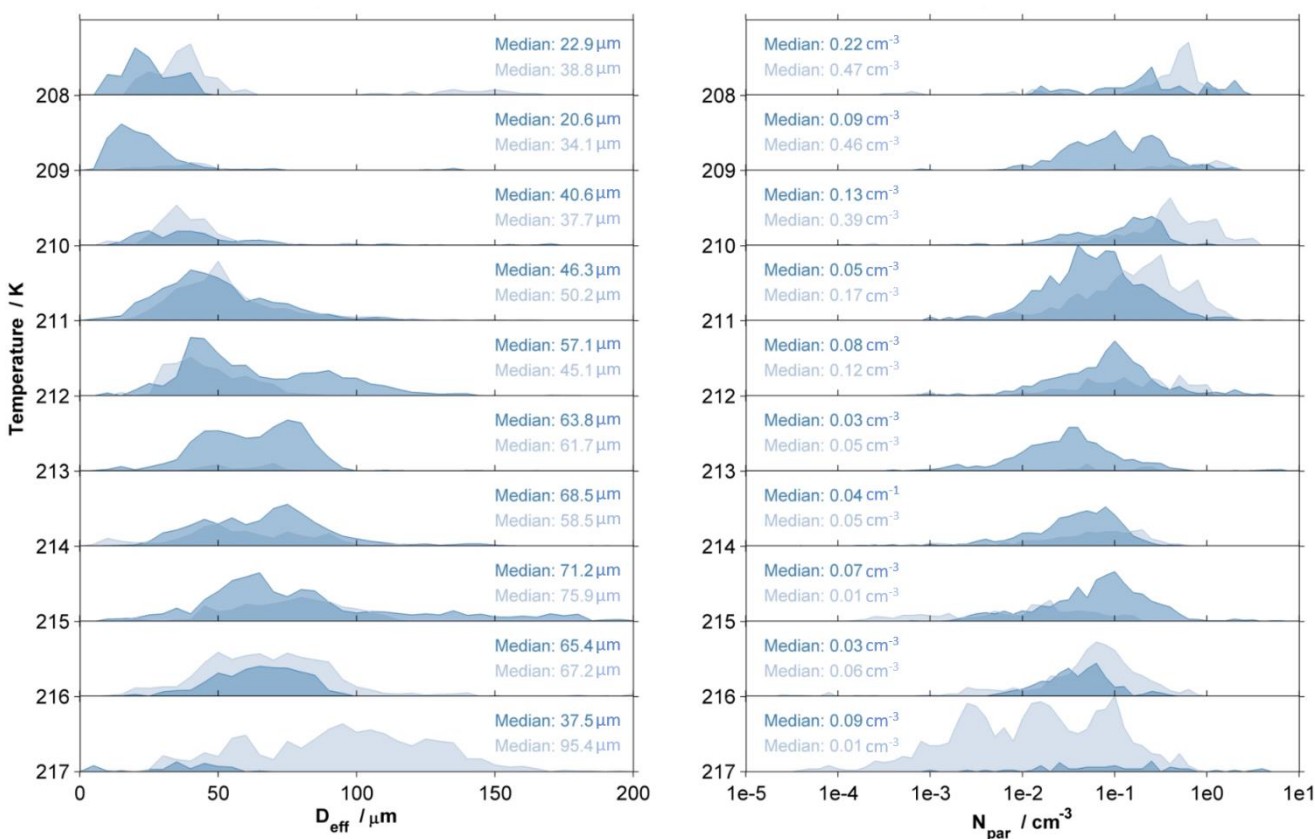

**Figure 8:** Temperature dependent relative distribution of the measured ice particle effective diameter (left) and ice number concentration (right) derived from CAS-DLR/CIPg-UniM and NIXE-CAPS for all cirrus clouds during ML-CIRRUS, where coordinated lidar and in-situ measurements were available (Table 1). The light blue color indicates measurements in low PLDR mode clouds; the one in dark blue show measurements for high PLDR mode clouds.