# Peer review of "Investigating an indirect aviation effect on mid-latitude cirrus clouds - linking lidar derived optical properties to in-situ measurements"

_Atmospheric Chemistry and Physics, 2022_

## Author Comment (AC1)

The authors use a data set of airborne lidar and in-situ measurements to study the effect of aviation on the optical and microphysical properties of natural cirrus clouds. In an earlier study by some of the same authors, lidar measurements of the particle linear depolarization ratio (PLDR) were used together with backward trajectories to identify cirrus clouds that likely formed in regions of high and low air-traffic density. The present work follows up on the previous study by adding the analysis of coinciding in-situ measurements of the ice crystal size distribution and the ice crystal number concentration (ICNC) related to the two PLDR modes to the investigation of an indirect aviation effect on mid-latitude cirrus clouds. The authors conclude that cirrus clouds that are affected by aviation as indicated by higher values of PLDR also show larger effective ice crystal radii and lower ICNC compared to unperturbed, low-PLDR cirrus.

The authors address an interesting topic that is certainly relevant to the readers of ACP. They have a unique set of airborne observations at their disposal. However, the study itself as well as the presentation of the results need considerable improvement before they can be accepted for publication in ACP. Below is a list of major and minor comments to the authors.

***We thank this reviewer for the helpful suggestions to improve our manuscript! The answers to the reviewer's comments will be given in italic and bold.***

Major comments

- The text is quite repetitive, often imprecise, and sometimes just confused. Please refer to the minor comments below for details. The authors should review the text carefully. Language editing is urgently needed before publication.

***We have carefully revised the text following the suggestions.***

- The description of the data set and methodology is rather sloppy:
  - What is the procedure for ensuring that lidar and in-situ instruments have probed the same cloud? How have sections of the flights during ML-CIRRUS been selected for the analysis presented here?

  ***We added the following paragraph to explain, how the flights during ML-Cirrus have been selected for this study:***

  ***'However, only eight of the 16 flights were designed in a way, that they provide coordinated lidar and in-situ measurements. The sampling strategy during these flights were as follows: First, the HALO aircraft flew at higher altitudes for sounding the cirrus clouds with lidar (lidar leg). Subsequently, the cirrus clouds were probed by in-situ measurements at several flight altitudes within the cirrus clouds (in-situ leg). […] For our study only these eight flights with coordinated lidar and in-situ measurements are relevant. Information on the flights (including their Mission ID to make it comparable to Urbanek et al. (2018)) are given in Table 1.'***

  - Does the data set include all cases of coincident lidar and in-situ measurements? What is the volume of the data set of coincident lidar and in-situ observations? Is it a few minutes or several hours? This information is also missing in the corresponding plots.

*We used all coincident lidar and in-situ measurements for this study. And added the following sentence to provide information of the duration of the measurements:*

> *'Typical lidar legs took about 30 min to 50 min; with a typical aircraft speed of 200 m/s that result in an observed cloud dimension of about 360 km to 600 km. The in-situ legs took a minimum of 10 min per constant flight altitude.'*

o   Instead of providing information on the water vapour measurements (which are not used at all in this study), the authors could define the backscatter ratio for lidar non-experts or state what typical cirrus PLDR values are.

*We removed the information on the water vapor measurements and included information on the backscatter ratio and the PLDR.*

o   How are regions of high and low air-traffic density defined and how is the connection made to the measurements? You could provide a quick review of the procedure in Urbanek et al. (2018) and clarify that your analysis is the continuation of their work based on the same cases. If this is clear, you wouldn't need to reproduce an already published figure that isn't really necessary here (as is obvious from the fact that it isn't discussed at all). In any case, figures shouldn't be reproduced from an earlier publication.

*We followed this suggestion and clarified that this analysis is the continuation of the work by Urbanek et al. (2018). We thus removed the figure that was already published in this former study. Furthermore, we included a short review of the procedure by Urbanek et al. (2018) to define high and low air-traffic density:*

*'Urbanek et al. (2018) grouped these flights respectively if the cirrus clouds developed in regions with enhanced background aerosols due to aviation or in rather pristine regions. Therefore, they used 24-hr backward trajectories calculated with the trajectory module of CLaMS (McKenna et al., 2002). They used the maximum cloud ice water content to determine the most probable location of the cirrus development and compared that to maps of enhanced background aerosols due to aircraft emissions (Stettler et al., 2013).'*

o   The section on the in-situ measurements is quite confusing and ends with a statement that comparison of the data sets shows good agreement. While we don't know what that means or how the data of the different instruments have been combined, it seems that the consideration of data from NIXE-CAPS is sufficient for the purpose of this study. Please stick to the necessary data to keep the study simple.

*We rearranged the text in the manuscript to emphasize and clarify the use of the different cloud probes. We used mainly the combination of the CAS, CIP and PIP for the data evaluation. We also added a sentence that the data of the combination of the CAS, CIP and PIP and the NIXE-CAPS are redundant.*

o   The analysis of cirrus measurements for different temperatures could already be motivated and outlined in the methodology section. The availability of in-situ measurements of temperature and humidity is first mentioned in line 168. Why are these measurements and the corresponding instruments not listed in the data section?

*We now include these measurements already in the data section.*

- The authors should revise the presentation of their data and results:
  - Table 1 is a reproduction of parts of Table 3 in Voigt et al. (2017). It would be more useful for the reader if the table was to include information on the inferred parameters for different cases rather than generic mission information. I would like to see, for instance, number of data points, length of measurements, mean PLDR, mean ICNC, mean Deff, etc. For readers of a scientific paper, dates – as used later in the plots – are much more tangible than arbitrary mission IDs.

*The reviewer is right, that the Table in the current version does not provide enough information. We included the median PLDR as given in our previous study (Urbanek et al., 2018) and added information on the temperature of the in-situ legs. However, we could not give mean ICNC and mean Deff as we did not look for that for specific clouds but for specific temperatures. We thus added information on the number of data points for the temperature steps in Figure 8.*

  - Figure 1 is a reproduction of Urbanek et al. (2018). Why not just state that their cases will be used for closer inspection in this study? In any case, the authors should briefly review the approach in Urbanek et al. (2018) for allocating high and low PLDR cases to regions of high and low air-traffic density, respectively.

*We removed Figure 1 and clearly identify that this is a follow-on study. This also includes a brief review of the approach used by Urbanek et al. (2018) (see comments above).*

  - It is not clear what the readers should take from the lidar plots in Figures 2 and 5. It would be more straightforward to combine Figures 2c and 2d with Figure 3 (and analogous Figures 5c and d with Figure 6) into key plots that present all frequency distributions. In any case, please choose a plot mode that allows the readers to separate the different cases. Curves without fill would be an obvious choice over overlaid bar plots. Please also add information on the data sets to the plots if you decide against presenting them in a table.

*Thanks for this comment, but we want to keep Figures 2 and 5 in the current form. However, we follow the advice and modified the overlay plots.*

  - Figure 4 could be omitted. Its content is fully described in a few sentences. Instead, it would be nice to get more information about the cirrus classification scheme either in the discussion of the second case study or already in the methodology section.

*We removed Figure 4 but included a brief description of the classification scheme.*

  - I suggest to motivate the analysis of clouds at different temperature already in the methodology. Figure 8 could be discussed before Figure 7. In addition, the left panel in Figure 7 seems unnecessary in light of Figure 8 and its discussion. Stating the message of that figure in the text is sufficient. It would be very helpful if median values were also marked in the plots in Figure 8, for instance by vertical lines.

*We followed this suggestion and removed the current left panel of Figure 7 and included information on the median values. However, we included a new panel*

*showing the comparison of the temperature range where in-situ measurements of both cloud-classes is available.*

- The data show differences in ICNC and Deff for clouds with different PLDR. The value of this finding would be much increased if the authors where to present the result of a significance test.

*Thanks for this advice. We now include the results of a significance test in the supplement.*

Minor comments

- line 36: Bräuer et al. (2021a) doesn't seem to be related to effects of biofuel. Please remove.

*Done*

- line 38: An increase in ICNC is also found in Marjani et al. (2022): https://doi.org/10.1029/2021GL096173.

*We included this reference.*

- line 40: IN should be INP as introduced in line 275.

*Done*

- line 53: Redundant. This has just been clarified in the previous paragraph.

*Done*

- line 55: ML-Cirrus or ML-CIRRUS?

**Corrected**

- line 57: pristine really means air-traffic free or with low air-traffic density. Please clarify throughout the manuscript.

*Done*

- line 64: less cirrus formation? Do you mean a reduction in cirrus occurrence?

*Yes, we corrected that.*

- line 69: energy forcing = radiative forcing?

*In their publication they differentiate between radiative forcing and energy forcing. For the contrails they refer to energy forcing.*

- line 75: They also found... Not clear what is stated here.

*We modified this sentence.*

- line 151-156: This introduction to the first case would be more trustworthy if the authors had provided a review of Urbanek et al. (2018) earlier in the manuscript. Please provide link to Urbanek (2018).

*We included review of Urbanek et al. (2018) earlier in the revised manuscript.*

- line 160-164: Consider putting this into a table.

*The median values of the PLDR are now given in Table 1.*

- line 164: It is not clear how embedded contrails have been identified.

*The embedded contrails have been identified using the CoCiP (Shumann 2012) calculations. We only rely on these calculations and did not include measurements to identify the embedded contrails.*

- line 170-173: As raised in the major comments, this statement indicates that data from NIXE-CAPS should be sufficient for the purpose of this study then.

*Thanks for pointing out, that is point has not been very clearly explained. The study was conducted using the data from the combination of the CAS, CIPg-UniM and the PIP. The probe combination offers a measurement size range between 3 and 6400 $\mu$m. Deff never exceeded a value of 200 µm, therefore only the CAS and CIPg-UniM were the main instruments feeding into the analysis. This combination has been used in various other publications (Righi et al., 2020, Wang et al., 2023). We added the NIXE CAPS data (with the same size range) for one flight where no data from the main-probe-combination was available due to a failure of the CIPg-UniM. The data sets agreed well when compared for selected flights. We clarified this in the manuscript*

line 175: Significant digits: give either exact numbers or rounded values, but not a combination of both.

*Corrected*

- line 183-186: Please revise the description for precision. Is it a cloud or a flow? What's the dimension related to several kilometres?

*We corrected this.*

- line 199-205: Consider putting this into a table.

*The median values of the PLDR are now given in Table 1.*

- line 254: This cannot be seen in the current Table 1. It could be, if the authors where to include a table that presents the values of the considered parameters for the different cases.

*We revised Table 1.*

- line 303: It would be nice to list these open questions as motivation for further research

*We included one of the most urgent question.*

---

## Author Comment (AC2)

The authors investigate collocated in situ and lidar observations for two classes of high and low particulate linear depolarization ratio (PLDR) measured by the WALES lidar during the ML-CIRRUS field campaign as identified by Urbanek et al. (2018). The in situ measurements of interest are ice effective diameter and ice number concentration, and measurements in high PLDR (formed in air traffic regions) and low PLDR (formed in "pristine" regions) cirrus clouds are compared. After a quick presentation of the ML-CIRRUS campaign, of the WALES lidar system, and of the in situ instrumentation, comparisons are shown for two contrail cirrus clouds, two warm conveyor belt cirrus clouds, and finally for all flights, for which the comparisons are shown at 10 temperatures between 208 and 217 K. The authors conclude that in the 210-215 K temperature range, chosen to have a "sufficient contribution of both cloud types", high PLDR mode clouds have larger effective diameter and lower number concentration, which is "an indication for more heterogeneous freezing due to aviation induces emissions".

Even though not clarified in the abstract and in the introduction, this paper is an extension of the work presented by Urbanek et al. (2018), who found 2 classes of cirrus clouds with higher and lower PLDR, and who showed that they formed in busy air traffic regions and in regions with low aviation emissions, respectively. Urbanek et al. (2018) stated that heterogeneous freezing on emitted exhaust particles could explain the lower super saturations and higher PLDR that they found.

The lidar results derived from Urbanek et al. (2018) are well presented, which is convenient for the reader, but I was expecting to see more solid material about the in situ measurements. Section 2.3 about the in situ instrumentation does not discuss the expected performances of the instruments. It is not clear if NIXE-CAPS alone is sufficient for this study. I was expecting detailed presentations and discussions of the "combined" and "coordinated" lidar and in situ measurements, with discussions regarding the spatial and temporal collocations of the in situ and lidar legs. In the two case studies, the authors compare cirrus clouds of the same type and discuss the relevance of the comparisons. No conclusion could be drawn for the second case study (sect. 3.2) because temperatures differed by about 10 K (223 K and 232 K). In section 3.3 where all flights are combined for the comparisons, the relevance of the comparisons is not discussed. The authors present comparisons vs. temperature (Fig. 8), but only between 208 and 217 K (10 temperature values). As a matter of fact, not all in situ data were used. The authors need to detail which flights were selected for Fig. 8, why, how many PSDs per temperature range in the high and low PLDR modes, etc… I am not convinced that different number of samples justifies ignoring the cases which do not match the expectations. This might indicate that other phenomena come into play. I understand that such comparisons are challenging and that the campaign was not designed for this type of analysis.

In my opinion, the analyses presented in the manuscript are incomplete, and this manuscript does not represent a sufficient contribution to scientific progress to be accepted for publication in ACP. However, the scientific question is important, and perhaps the following suggestions and questions will help the authors pursuing this effort.

***We thank this reviewer for the careful reading and the helpful suggestions to improve the manuscript! The answers to the reviewer's comments are given in italic and bold.***

**Specific comments**

Abstract:

1. Lines 20-22: this is misleading. These findings were published by Urbanek et al. (2018).

   ***We now clearly indicate already in the Abstract that this study is a follow-on study of the work started by Urbanek et al. (2018).***

2. Lines 22-23: in my opinion, this is an overstatement.

   ***We skipped that sentence.***

Introduction:

3. Line 83: I strongly suggest to clarify that the cirrus clouds formed in air traffic regions or in pristine regions are identified according to the classification established by Urbanek et al. (2018) which is based on lidar measurements of PLDR. Lines 55 to 59 could be moved here.

   ***We now mention clearly that this is a work that extends the study by Urbanek et al. (2018) and that the same clouds as in this former study are investigated.***

Section 2 method

Section 2.1:

4. Only 8 flights are listed in Table 1 (which should be introduced in the text), whereas 16 flights are shown in Fig. 1, with combined remote sensing and in situ observations for all these flights if I understand the text correctly. Please confirm that only the 8 flights listed in Table 1 are relevant for this study and explain why. For clarity, only the 8 flights listed in Table 1 should be shown in Fig. 1.

   ***This is right, we did not explain why only those eight flights are uses. We corrected this in the revised manuscript and also explain better the measurement strategy for the collocated lidar and in-situ measurements by adding the following paragraph:***

   ***'However, only eight of the 16 flights were designed in a way, that they provide coordinated lidar and in-situ measurements. The sampling strategy during these flights were as follows: First, the HALO aircraft flew at higher altitudes for sounding the cirrus clouds with lidar. Subsequently, the cirrus clouds were probed by in-situ measurements at several flight altitudes within the cirrus clouds. For our study only these eight flights with coordinated lidar and in-situ measurements are relevant. Information on the flights (including their Mission ID to make it comparable to Urbanek et al. (2018)) are given in Table 1.'***

5. Later in the paper, the authors refer to Table 1 when discussing number of observations in various conditions. I suggest providing for each flight information such as the number of PSDs, temperature and altitude range, PLDR range, etc…A suggestion is to add a dedicated table at the beginning of Section 3.

*We like this suggestion and modified Table 1; we included information on the median PLDR (as given in Urbanek et al., 2018), of the altitude range of the cloud and the temperature of the in-situ measurements within the cloud (that was observed also by lidar). However, we did not include information on the PSDs as we did not perform the comparisons on a cloud by cloud base but a temperature base.*

6. Please define the PLDR: ratio of which quantities?

*Done*

Section 2.3

7. How many instruments were involved for this study? How do NIXE-CAPS, CAS-DLR and CIPg-UniM compare in terms of sensitivity range? Which instrument(s) was/were not available when only data from the NIXE-CAPS instrument were available and what are the possible consequences for this study? Please specify when you state that "comparison of the data sets for all other days showed a good agreement".

*The study was conducted using the data from the combination of the CAS, CIPg-UniM and the PIP. The probe combination offers a measurement size range between 3 and 6400 μm. Deff never exceeded a value of 200 μm, therefore only the CAS and CIPg-UniM were the main instruments feeding into the analysis. This combination has been used in various other publications (Righi et al., 2020, Wang et al., 2023). We added the NIXE CAPS data (with the same size range) for one flight where no data from the main-probe-combination was available due to a failure of the CIPg-UniM. The data sets agreed well when compared for selected flights. We clarified this in the manuscript.*

8. Please define the effective diameter Deff and explain how it is computed from the PSDs. I anticipate that assumptions are necessary. I could not find the Schumann et al. (2010) reference.

*The reference was change to Schumann et al. 2011. Please excuse the confusion.*

*Schumann, U.; Mayer, B.; Gierens, K.; Unterstrasser, S.; Jessberger, P.; Petzold, A.; Voigt, C. & Gayet, J.-F. Effective Radius of Ice Particles in Cirrus and Contrails, Journal of the Atmospheric Sciences, American Meteorological Society, 2011, 68, 300 - 321*

9. Line 139: the data are averaged over 5 s intervals. Can you comment on the number of PSDs for each flight? This piece of information should be provided.

*The number of data points n for Npart and Deff and each temperature interval are mentioned in Fig. 8, next to the median. Each data point n refers to a 5 s interval, where one averaged PSD in that time segment was used to derive N and Deff.*

Section 3 results

10. It might be worth clarifying or reminding that the goal is to compare in situ measurements in two classes of cirrus clouds exhibiting large and low PLDR as established by Urbanek et al. (2018). The authors actually investigate the microphysical properties.

***We have now added the following sentences to clarify that we extend the former study by comparing in-situ measurements:***

***'In a previous study (Urbanek et al., 2018) we found, that cirrus clouds evolved in regions with enhanced air traffic show larger mean values of the PLDR than cirrus clouds forming in rather pristine regions. In this study we extend the investigation of an impact of aviation on the microphysical properties of the cirrus clouds by comparing in-situ measurements performed within these two cloud classes (of high and low PLDR).'***

Section 3.1

11. Can you explain why the cirrus with embedded fresh contrails observed on April 7th, 2014 is not somewhat affected by aviation exhaust. I could not have access to Stettler et al. (2013), but nevertheless, I think that this deserves an explanation in this paper.

***We included this information in Section 2.1.***

12. Line 171: why care about a missing instrument on 7 March? Unless you meant 7 April? Why no impact on the results if an instrument is missing?

***This was a typo. We meant 7 April.***

13. Is it possible to point to the regions with embedded fresh contrails in the lidar plots? Are these regions identified from observations or from the model?

***The regions with embedded contrails were identified using the CoCIP Model (Schumann et al., 2012).***

14. The authors should present in details the spatial and temporal collocations of the lidar measurements and in situ measurements presented in this paper. I note that in situ measurements are shown in Wang et al. (2022) for the 26 March, 2014 flight. The authors should clarify which in situ legs are used in this work, and how they were chosen. The number of PSDs for each flight, altitudes and temperatures could be added in Figure 3 for clarity.

***We added information on the temperature of the in-situ flight legs, that was used for our comparison, in Table 1. For most of the comparison we did not restrict ourselves to only one flight leg (at one temperature) but used all the available co-located information of lidar and in-situ measurements. For the comparison of the contrail cirrus, we are in the good position, that the in-situ flight tracks were performed at the same temperature levels.***

15. I see large Deff > 100 um on April 7th even though median Deff is smaller than on March 26th. This should be acknowledged. Authors should discuss in this section

the various possible reasons for the larger N on April 7th compared to March 26th. For instance, it seems that the in situ measurements were higher in the cloud on April 7th, and perhaps closer to an embedded fresh contrail? Plots showing N vs. Deff for each case could be useful for this discussion.

*The large values were due to the sampling on the base of the cloud; partly inside and partly outside. A closer look at this in-situ leg, showed, that it is not reasonable to use this flight leg as in in-situ in-cloud measurement. Thus, we better filtered our data and included a new version of the analysis. We also did a better filtering for the second case study with respect to collocated measurements.*

Section 3.2

16. Lines 183-186: are you describing liquid origin cirrus clouds (e.g. Luebke et al., ACP, 2016)?

*Yes, warm conveyor belts are a type of liquid origin cirrus, but at their top they can be connected to in-situ formed ice clouds.*

17. The authors should present in details the spatial and temporal collocations of the lidar and in situ measurements. Same comments as for the previous case study.

*We added information on the co-location of the lidar and in-situ measurements in the general description of our study (Section 2).*

*'However, only eight of the 16 flights were designed in a way, that they provide coordinated lidar and in-situ measurements. The sampling strategy during these flights were as follows: First, the HALO aircraft flew at higher altitudes for sounding the cirrus clouds with lidar (lidar leg). Subsequently, the cirrus clouds were probed by in-situ measurements at several flight altitudes within the cirrus clouds (in-situ leg). Typical lidar legs took about 30 min to 50 min; with a typical aircraft speed of 200 m/s that result in an observed cloud dimension of about 360 km to 600 km. The in-situ legs took a minimum of 10 min per constant flight altitude. For our study only these eight flights with coordinated lidar and in-situ measurements are relevant.'*

Section 3.3

18. I believe that Fig. 7 is not really useful here and that Fig. 8 is the most interesting. That being said, the warmest temperature is 217 K, which indicates that the comparisons presented in section 3.2 (223 and 232 K) are not included. I really do not understand. I see that Fig. 8 uses data from CAS-DLR/CIPg-UniM and NIXE-CAPS, but having only NIXE-CAPS did not seem to be an issue for the case study presented in section 3.1. Please justify this choice and detail which flights were used to create Fig. 8 and which cirrus types. I suggest to give the number of PSDs at each temperature for the high and low PLDR range to avoid vague discussions.

*The reviewer is right, the left part of Figure 7 is not of any value, so we removed it. However, we believe that the right part provides useful information. As we now also included the temperature in which the in-situ measurements were performed in Table 1 it gets more obvious why we restricted our comparison in Figure 8 only to the temperature range between 208 K and*

*217 K. For the warmer temperatures we have either only measurements in a 'high PLDR cloud' or in a 'low PLDR cloud'.*

19. I see Deff larger in the high PLDR mode (dark blue) than in the low PLDR mode (light blue) only at 210 K and between 212 and 214 K. N is smaller in the high PLDR mode except at 215 K and 217 K. I am not convinced that the different number of samples justifies ignoring the cases which do not match the expectations. This might indicate that other phenomena come into play.

*This might have been misleading in the text. We did not omit the cases which did not match our expectations, we omitted the temperature ranges, were no comparison was possible due to a highly imbalanced number of available datapoints. We now include the number of datapoints in Figure 8. To our understanding an imbalanced number of datapoints for both cloud types provides no significant comparison and are thus omitted in the general view of Figure 7.*

20. Line 247: The "tendency towards larger Deff with temperatures" is consistent with numerous publications found in the literature, which should be cited.

*Done*

Discussion and conclusions (should be section 4?):

21. Lines 265-266: respectfully, I think that this is an overstatement. I see these findings only at 210 K and between 212 and 214 K.

*We removed this sentence and tried to show our indications more carefully.*

---

## Author Response (AR2)

**We thank the reviewers again for the careful reading and the suggestions that helped us improve our manuscript. The answers to the reviewers will be given after each comment in bold font.**

**Reply to Referee #1**

The authors have addressed some of the concerns but major questions still remain. In addition, the presentation of the work could be further improved.

 - I would suggest to point out that the effect investigated here falls under what is also known as non-CO2 effects of aviation.

**We included this in the introduction.**

- The abstract reads more like an introduction. Please revise to provide all the major findings, i.e. specific results, of your study.

**We include now the major findings of this study and modified the abstract accordingly:** *"… In-situ measurements for both cloud types (high mode PLDR -aviation effected- and low mode PLDR - pristine- cirrus) can be reliably compared in a temperature range between 210 K and 215 K.  Within this temperature range we find that high mode PLDR cirrus clouds tend to show larger effective ice particle diameters with a median value of 61.4 µm compared to 50.7 µm for low mode PLDR pristine cirrus clouds. Larger effective ice particles in aviation influenced (high mode PLDR) cirrus are connected to lower ice particle number concentration with a median value of 0.05 cm$^{-3}$ compared to 0.11 cm$^{-3}$ (low mode PLDR), which evolved in more pristine regions with only little impact from aviation. We suspect that a suppression of homogeneous ice formation by the heterogeneously freezing soot aerosol particles included in the areas affected by air traffic is the cause of the reduced ice crystal concentrations."* **. However, we decided to not change the first part of the abstract as it was an outcome from the first review process to make a direct link to the study of Urbanek et al., 2018 already in the abstract.**

- INP is introduced for what should be ice nucleating particles but later in the text the authors still use IN.

**We changed that and use only INP in the revised version.**

- Please double-check the list of references. Some names are spelled incorrectly (e.g. Majani, line 38; Theo, line 67)

**Thank you! We checked and corrected that.**

- line 72: Please be specific as to how much of the AOD decline can be attributed to air-traffic reductions and regarding the amount of change to the radiative forcing.

**We modified the paragraph accordingly:** *'An integrated study, using aircraft, satellite and modelling data, showed a reduction of the aerosol optical depth (AOD) over Europe in May 2020 (Voigt et al., 2022). Although it is not clear whether the decrease in AOD was caused mainly by anthropogenic or meteorological influences, the authors suggest, that it was partly caused by the 80% decline in air traffic, as the aerosol number concentration decreased at flight altitudes compared to the reference years. In addition, comparisons of the measured black carbon mass concentration with model results from EMAC indicated a 40% reduction related to lockdown effects*

*(Krüger et al., 2022). The reduction in air traffic over Europe furthermore led to a reduction in contrail cover and as a consequence in radiative forcing from contrails. Contrail radiative forcing was calculated with the contrail cirrus prediction model (CoCiP; Schumann 2012) for April 16, 2020 for two scenarios; using air traffic data from 2019 and for 2020. For the same meteorology, the simulated contrail radiative forcing decreases by about 80% for the reduced air traffic in 2020 compared to 2019.'*

- line 107: development or formation?

**We changed that to 'formation'.**

- While I like colourful lidar plots as much as anybody, I cannot see the added value of the ones on Figures 1 and 3. The reply letter also gives no justification why those plots are needed.

**We prefer to keep these lidar plots, as they nicely show the distribution of the PLDR values within the cloud and, especially, that the PLDR does not show any altitude or time separation.**

- The two case studies can be presented more comprehensively by combining the frequency distributions of Figure 1c and d (3c and d) with those in Figure 2 (4) into a single figure with a homogenized design of the frequency distributions.

**As we prefer to keep the lidar plots, we also prefer to keep the distribution of the PLDR together with the lidar plots instead of combining them with the frequency distribution of the in-situ measurements.**

- line 198: measurement = meteorological?

**We changed that to 'meteorological'.**

- line 198 - 201: Redundant, this is now stated in the section on data and methods.

**We removed that.**

- line 220 and following: This should be part of the methods section.

**We moved that to the methods section.**

- It seems that measurements of RHi are available. Why are these not used in the results section? Temperature alone doesn't seem to cover the full picture. This becomes particularly clear in the discussion when the authors argue that RHi can provide information regarding the likelihood of homogeneous and heterogeneous nucleation.

**We did not include measurements and analysis of RHi in this manuscript, as a detailed discussion on RHi distribution for high mode and low mode PLDR clouds was already included in the publication by Urbanek et al., 2018.**

- The authors make an argument, that the suitable range for comparison is between 210 and 215 K. Therefore, I don't see the need to show plots for data outside that temperature range. The authors should show only data for the temperature range from 210 to 215 K, i.e. omit the left column of

Figure 5 and the corresponding lines in Figure 6. The conclusion of using all data can be provided in the text.

**We followed that advice and changed that.**

**Reply to Referee #2**

The vast majority of my initial comments have been addressed in the revised manuscript, which is significantly improved.

My main comment is that both the abstract and the conclusion should reflect that in situ measurements for both types of clouds (high and low PLDR) could be reliably compared between 210-215 K and that clouds with higher PLDR "tend to" show larger mean effective ice particle diameters connected to smaller ice particle number concentration than the cirrus clouds with lower PLDR.

**We added the temperature range in which in-situ measurements for both cloud types can be reliably compared in the abstract and in the conclusion. Further, we discuss in more detail the differences between the high PLDR, aviation affected, and low PLDR, pristine cirrus clouds, in the abstract.**

Other comments
Line 58: do you confirm that "The measurement study by Urbanek et al. (1998)" should be replaced with "The measurement study by Urbanek et al. (2018)"?

**Yes! We corrected that.**

Line 69: this recent publication could be cited: Duda, D. P., Smith, W. L., Bedka, S., Spangenberg, D., Chee, T., & Minnis, P. (2023). Impact of COVID-19-related air traffic reductions on the coverage and radiative effects of linear persistent contrails over conterminous United States and surrounding oceanic routes. Journal of Geophysical Research: Atmospheres, 128, e2022JD037554. https://doi.org/10.1029/2022JD037554

**We added this publication.**

Line 145 and line 199: I see again that the authors mention missing CIP-UniM data on 7 March 2014 whereas it seems that it was on 7 April 2014.

**Sorry! We changed that.**

Line 184: "26 April" should be "26 March" I believe. Median PLDR = 0.52 but is 0.51 in Table 1. Please explain this difference or correct.

**We changed that.**

Line 185: I see in the text that the mode of the PLDR distribution = 0.34 and median = 0.29, but looking at the plot, it seems that both values are actually larger, perhaps 0.44 and 0.39, respectively. Median PLDR in Table 1 is 0.39, which seems correct. Can you verify?

**You are right, thank you for pointing this out. We corrected the values in the text.**

Line 251: "…mean value of 225 K for the flight track of 11 April". I see 226-227 K in Table 1. Can you clarify?

**We corrected that.**

Lines 268 to 274: it seems that these results are for all flights, because the temperature range is from 206 to 238 K, whereas temperature range in Table 1 is from 208 to 227 K. Please clarify. Perhaps these first sentences are not relevant anymore.

**You are right, these first sentences are not relevant anymore. We removed them.**

Line 307: looking at Fig. 6 and at the values next to the distributions, it seems that "The corresponding values for the low mode PLDR clouds are 28.8 μm and 67.2 μm, respectively" should read "The corresponding values for the low mode PLDR clouds are 38.8 μm and 67.2 μm, respectively".

**We corrected that.**

Lines 315-318: I do not see light blue Npar (low PLDR) larger than dark blue Npar (high PLDR) at all temperatures. What are "all considered temperature ranges"? At 208 K, I see indeed 0.48 cm-1 for the low mode PLDR cirrus but 0.23 cm-1 for the high mode PLDR cirrus.

**Following the advice of the second reviewer we now include only the temperature range from 210-215 K in our comparison. However, we made clear in the text following Figure 6, that Npar (low PLDR) is not larger than the corresponding Npar (high PLDR) for all temperature steps.**

Lines 318-320: I suggest to be more specific and to modify the sentence, which could read: "However, as can be seen from the number of datapoints for each comparison given in Fig.6, these last results have to be treated with care, as we tend to have a larger number of high mode cirrus cloud cases in the lower temperature ranges (except at 208 K and 210 K) and a dominance of low mode cirrus cloud cases in the higher temperature range, namely at 216 and 217 K.

**We followed your advice and changed the sentence accordingly.**

Line 330: I strongly suggest specifying that this temperature range with sufficient contribution of both cloud types is 210-215 K. I also think that a more accurate statement would read for instance: "the high PLDR mode clouds tend to show lower ice particle number concentrations with larger effective diameters compared to low PLDR mode clouds".

**We followed your advice and changed the statement accordingly.**

**Technical comments**
Line 20: PLRD => PLDR **- Done**
Line 23: PLRD => PLDR **- Done**
Line 262: "Unfortunately, now lidar measurements…" => "Unfortunately, no lidar measurements…" **- Done**
Line 284: "as now significant comparison is possible" => "as no significant comparison is possible" **- Done**